



**Impact of the COVID-19 pandemic on the observed vertical distributions of**
**PM$_{2.5}$, NO$_x$, and O$_3$ from a tower in the Pearl River Delta**
Lei Li[a, b, c], Chao Lu[d], Pak-Wai Chan[e], Zi-Juan Lan[f], Wen-Hai Zhang[g], Hong-Long Yang[d] and Hai-Chao Wang[a, b, c]*
a School of Atmospheric Sciences, Sun Yat-Sen University, Zhuhai, 519082, PR China
b Guangdong Provincial Observation and Research Station for Climate Environment and Air Quality Change in the
Pearl River Estuary, Zhuhai, 519082, China
c Key Laboratory of Tropical Atmosphere-Ocean System (Sun Yat-sen University), Ministry of Education, Zhuhai,
519082, China
d Shenzhen National Climate Observatory, Meteorological Bureau of Shenzhen Municipality, Shenzhen, 518040,
PR China
e Hong Kong Observatory, 999077, Hong Kong
f Shenzhen Research Academy of Environmental Sciences, Shenzhen, 518001, PR China
g Shenzhen Academy of Severe Storms Science, Shenzhen, 518057, PR China
Corresponding author: wanghch27@mail.sysu.edu.cn
**Abstract.** The outbreak of the 2019 novel coronavirus (COVID-19) has brought tremendous impact and influence
on human health and social economy around the world. The lockdown implemented in China, starting on 23
January 2020, led to large reductions in human activities and the associated emissions. Sharp declines in primary
pollution provided a unique chance to examine the relationships between anthropogenic emissions and air quality.
Here, we report measurements of air pollutants and meteorological parameters at different heights on a tall tower in
the Pearl River Delta, China, to investigate the response of the vertical scales of pollutants to reductions in human
activities. Compared to the pre-lockdown period (starting from 16 December 2019), the observations showed that





surface layer $NO_x$, $PM_{2.5}$ and mean values of the daily maximum 8 h average $O_3$ ($MDA8O_3$) had significant
reductions of 76.8%, 49.4%, and 18.6% respectively, but the average $O_3$ increased (9.7%) during lockdown period.
The vertical profiles of $NO_x$ and $O_3$ changed during the lockdown period, but not those of $PM_{2.5}$. The correlation
between $PM_{2.5}$ and $O_3$ was statistically significant, but not that between $PM_{2.5}$ and $NO_x$ for data collected at four
different heights during the lockdown period. The significance of these correlations was the opposite during the
pre-lockdown period, indicating that the main composition of $PM_{2.5}$ has changed dramatically since the lockdown,
which is transited from primary aerosol dominating or nitrate dominating (affected by $NO_x$) before lockdown to
secondary organic aerosol dominant dominating (affected by $O_3$) during the lockdown. We find weaker diurnal
variation of $O_3$ during the lockdown period is similar to the case at background regions. $O_3$ concentrations were not
sensitive to $NO_x$ concentrations during lockdown, which implies that $O_3$ levels during the lockdown are more
representative of the regional background, for which anthropogenic emissions are low and photochemical
formation is not a significant ozone source. This evidence suggests that significant reductions of anthropogenic
emissions are effective in simultaneous mitigation of $PM_{2.5}$ and $O_3$ levels.
Keywords: COVID-19 induced Lockdown, $PM_{2.5}$, $NO_x$, $O_3$, Tower Observation

## 1. Introduction

The coronavirus disease 2019 (COVID-19) pandemic has completely changed the world and caused great
losses of life globally. At present, over 200 countries and regions have been affected by the pandemic, and the
numbers of infections and deaths caused by severe acute respiratory syndrome coronavirus 2 (SARS-CoV-2) and its
variants are still rising (Wang et al., 2020). Many countries have chosen to implement lockdowns to bring the
pandemic under control; that is, to cut off the spread of SARS-CoV-2 by reducing gatherings and maintaining
social distancing among individuals. These measures have generally reduced human activity, decreasing or



completely halting manufacturing work and the movement of people. Although the lockdowns have had devastating
socioeconomic impacts, recent studies have shown them to be beneficial for the environment (Chakraborty and
Maity, 2020).

The reduction in human activities due to the pandemic has greatly decreased the emission of primary

pollutants. This, in turn, has caused significant impacts on regional air quality (Xing et al., 2020; Salma et al., 2020;
Wang et al., 2021; Kim et al., 2021) and even climate (Gettelman et al., 2021), albeit differing from region to
region. In South East Asia, the lockdown has led to a notable decrease in the aerosol optical depth over the region
and in pollution outflow over the oceanic areas, while a significant decrease (27%–30%) in tropospheric nitrogen
dioxide ($NO_2$) levels has been observed over territories not affected by seasonal biomass burning (Kanniah et al.,
2020). Srivastava (2020) noted that the aerosol optical depth had been reduced by up to 50% over the
Indo-Gangetic Plain during the lockdown period. In Italy, urban road traffic decreased by 48%–60% on average
during the country's periods of implemented lockdowns, which greatly decreased the concentrations of $NO_2$ and
particulate matter with aerodynamic diameter less than 10 μm ($PM_{10}$) and less than 2.5 μm ($PM_{2.5}$) (Gualtieri et al.,
2020). Rodríguez-Urrego& Rodríguez-Urrego (2020) found that the average $PM_{2.5}$ concentration of the 50 most
polluted capital cities in the world had decreased by 12% on average. By analysing the emissions data of 28 cities
in the USA during its first round of lockdowns (15 March 2020 to 25 April 2020), it was found that 2 out of 3 cities
showed greatly reduced $NO_2$ and carbon monoxide (CO) concentrations (with decreases up to 49% and 37%,
respectively) compared with the 2017–2019 historical baseline and pre-lockdown levels. These decreases in $NO_2$
and CO concentrations also increased in proportion to the local population density. However, the $PM_{2.5}$ and $PM_{10}$
concentrations only decreased significantly in north-eastern USA, California, and Nevada, which also recorded the
largest decreases in $NO_2$ concentrations (Rodríguez-Urrego& Rodríguez-Urrego, 2020).

China was the first country in the world to report SARS-CoV-2 infections to the World Health Organization.



The atmospheric environment of China was also significantly affected by the lockdown measures taken during the
pandemic, with some studies showing the atmospheric $NO_2$ concentrations to have been greatly reduced. These
reductions first occurred in Wuhan before spreading to the rest of China (Wang and Su, 2020). The Pearl River
Delta (PRD) is one of the most important economic zones in China and is also one of the most rapidly urbanising
regions in the world. The intensity of human activity in this region is also amongst the highest worldwide (Li et al.,
2021). The PRD was once severely affected by air pollution, which manifested as increasingly frequent haze
weather and rising PM concentrations. Because of the optimisation of industrial structures and implementation of
increasingly stringent pollution control measures, the air quality over the PRD had already been improved
significantly over the past decade (Zhang et al., 2015). Nonetheless, because the PRD contains immense
transportation networks and a dense distribution of factories, it has been difficult to stamp out pollutant emissions
completely. Therefore, the nitric oxide ($NO_x$), $PM_{2.5}$, and ozone ($O_3$) concentrations in the PRD often spike because
of unfavourable weather conditions (Li et al., 2020). The pandemic lockdowns have greatly reduced the intensity of
human activities in the PRD in a very short time, which has created a rare opportunity for the study of air pollution
mechanisms in the area.
Ever since the advent of the COVID-19 pandemic, numerous scholars have used this unique window of
opportunity to gain important insights into the mechanisms of air pollution. However, most of these studies were
based on ground-level data or space-based measurements of atmospheric column concentrations. By contrast, there
are no reports about the vertical distribution of air pollutants during the COVID-19 pandemic period. The vertical
distribution of air pollutants is a crucial piece of the puzzle for understanding how air pollution events are formed.
Meteorological towers are by far the most useful platforms for studies about the vertical distribution of near-surface
pollutants. Unlike tethered balloons or drones, meteorological platforms can be used to obtain continuous and
stable measurements over a long period of time. Numerous such studies have previously been performed using the





325-m-tall meteorological tower in Beijing (Meng et al., 2008; Sun et al., 2010; Sun et al., 2013) and the 300-m-tall
tower in Boulder, USA (Brown et al., 2013).

The PRD has one meteorological tower, the Shenzhen Meteorological Gradient Tower (SZMGT). The

monitoring equipment on this tower can be used to measure several air quality factors, including $PM_{2.5}$, $NO_x$, and
$O_3$ concentrations. Li et al. (2020) had analysed the vertical distribution of pollutants in the PRD during the peak
pollution season, based on air quality data and meteorological data obtained at the SZMGT from December 2017.
This provided useful insights about the vertical structure of air pollutant distribution in the PRD. Shenzhen is a very
developed city with active human activities (Li et al., 2015) and is facing the problem of air quality (Yang et al.,
2020). As the beginning of the COVID-19 pandemic coincided with the peak pollution season of the PRD, data on
the vertical distribution of pollutants recorded by the SZMGT during this period are invaluable for revealing how a
decrease in human activity may affect pollutant concentrations.

**2. Data and Methods**

The observational data, from 16 December 2019 to 15 February 2020, used in this study were from a

meteorological observation base on the east side of the Pearl River estuary; namely, the Shiyan Meteorological
Observation Base (hereinafter Shiyan Base), managed by the Shenzhen National Climate Observatory (Fig. 1a).
The base, which lies approximately 10 km from the coastline, is in the woodland area surrounding a reservoir.
Because the reservoir is an important source of drinking water for the population of Shenzhen, the environment
within 1 km around the SZMGT is protected by law and is rarely disturbed by human activity, ensuring that the
underlying surface will remain natural for a long time.

The entire area of Shenzhen is located within the subtropical monsoon climate zone. The dominant wind

direction in summer is south, and the airflow brings clean air from the sea to the base. In winter, the dominant wind
direction changes to a northerly one, and the airflow carries pollutants from the inland of the PRD to the base (Li et
al., 2020). The peak of the COVID-19 pandemic occurred mainly in winter, a period when the meteorological
conditions are generally unfavourable to the atmospheric environment of Shenzhen.

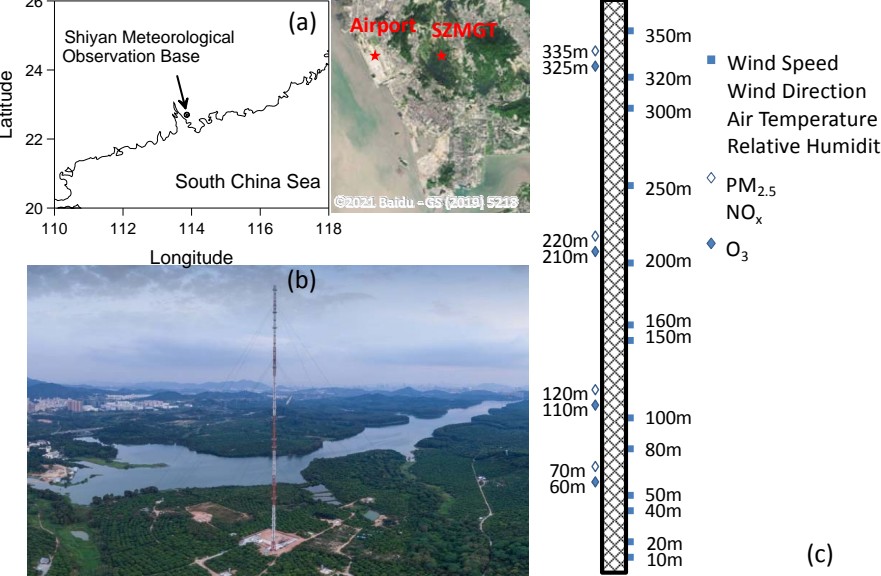

**Fig. 1.** Location of the Shiyan Meteorological Observation Base and Shenzhen meteorological gradient tower
(SZMGT): (a) Location of the Shiyan Meteorological Observation Base; (b) Arial view of the meteorological tower;
(c) Layout of the air quality and meteorological observation on the tower.

The SZMGT, which is 365 m tall (Fig. 1b), has 13 layers of meteorological observation platforms, starting

from 10 m up to 350 m (Fig. 1c). Four of those layers (i.e. at 60–70, 110–120, 210–220, and 325–335 m,
respectively) are atmospheric environmental observation platforms (Fig. 1c). The distance from the SZMGT to the
nearest built-up area is approximately 1 km. At 800 m north-east of the Shiyan Base, there is a busy highway from
which pollutants emitted by the vehicles passing through could influence the observational data on the tower. There
is also an airport located approximately 10 km west of the base which serves an estimated 356,000 flights in a


normal year. Thus, the airplanes taking off and landing at the airport also potentially influence the pollutant
concentration data recorded by the SZMGT (Li et al., 2020). An additional atmospheric environmental observation
station lies at the bottom of the SZMGT. Because this station is located on the ground, the height of its sampling
port is lower than that of the surrounding forest top.
The meteorological data used in the current study were collected at all 13 platform heights, as shown in Fig. 1c.
The environmental data were collected at the heights of 110–120, 210–220, and 325–335 m. The data at the height
of 60–70 m was not included in the analysis owing to the occurrence of equipment failure during the pandemic.
Data from the atmospheric environmental observation station at the bottom of the SZMGT were also used in the
current study.
The following are the equipment used at the SZMGT for sensing wind, temperature and humidity, and
visibility, respectively: the Vaisala WMT700 Ultrasonic Wind Sensor, Vaisala HMP155 Humidity and Temperature
Probe, and Vaisala PWD Present Weather Visibility Sensor. $PM_{2.5}$ concentration data are collected by Thermo
Scientific™ 5030i Sharp Particulate Monitoring equipment, $NO_x$ by the Thermo Scientific™ 42i Gas Analyzer, and
$O_3$ by the Thermo Scientific™ 49i Gas Analyzer. The data from the various instruments were downloaded at a
frequency of once every 5 minutes. Arithmetic averaging of the data was performed for all the elements, except for
the wind direction, to obtain hourly average data. The daily average data by arithmetic averaging were obtained
using the hourly average data over 24 hours. For determination of the wind direction, representative values were
obtained by calculating the highest wind frequency by the hour and by the day.

**3. Results and Discussion**
**3.1 Change of Pollutants Concentrations and Meteorological Elements**
Fig. 2 shows the daily mean concentrations of $PM_{2.5}$, $O_3$, and $NO_x$ observed at Shiyan Base in Shenzhen from



16 December 2019 to 15 February 2020, as well as the daily mean relative humidity (RH), daily mean temperature,
daily mean wind speed, and daily dominant wind direction of this period. Two key dates have been marked with
blue dotted lines on the $PM_{2.5}$, $O_3$, and $NO_x$ graphs: 15 January 2020 and 23 January 2020. The first case of
COVID-19 in Shenzhen was reported by local news outlets on 15 January 2020. The Shenzhen government reacted
very quickly to this news, despite the low number of patients with COVID-19 in the area at the time. The news was
immediately published on the official Shenzhen government website and social restriction measures were
implemented. On the advice of medical experts, a lockdown was imposed on Wuhan on January 23rd. The
Guangdong province, where Shenzhen is located, also activated its top-level emergency response on this day, and
all residents in Shenzhen and her neighbouring cities were instructed not to leave their homes unless necessary.
Therefore, after news about the COVID-19 pandemic first appeared on January 15th, the intensity of human
activity in Shenzhen (both manufacturing and traffic) began to decrease. By January 23rd, Shenzhen was virtually
shut down because of the strengthening of activity restrictions. Other than the most vitally important logistics
chains, very little traffic remained on the streets. Owing to a lack of data, it has not been possible to quantitatively
estimate the degree to which human activity decreased in Shenzhen during this period. Nonetheless, air traffic at
the airport west of Shiyan Base could provide some indication of the scale. In news reports, it was mentioned that
the number of passengers at the airport had decreased by as much as 79.49% in February 2020. Since February
usually coincides with the Spring Festival (Chinese New Year), this decrease in passenger volume is enough to
describe the magnitude by which human activity decreased in this region.
As shown in Figs. 2a and 2c, the daily mean concentrations of $PM_{2.5}$ and $NO_x$ closely tracked the
lockdown-mediated change in human activity. Since there were no cases of COVID-19 in Shenzhen before 15
January 2020, the local government did not impose any restrictions during the period between December 16th, 2019
and January 15th, 2020 and the pollutant concentrations remained high. After the first report of COVID-19 on



January 15th, many residents began to reduce the frequency of their outdoor activities owing to awareness of the
pandemic. Since these reductions in human activity were voluntary and not universal, the pollutant concentrations
only decreased slowly. However, the widespread implementation of high-level restrictions on January 23rd led to
drastic and sustained reductions in pollutant concentrations. The daily mean concentrations of $PM_{2.5}$ and $NO_x$
generally remained low after January 23rd, and their ranges of variation also became significantly narrower.
Although all three measured pollutants were reduced by the lockdown, the change in $NO_x$ was the most substantial.
This is because $NO_x$ is primarily derived from traffic emissions, and since the decrease in human activity also
decreased traffic emissions, the concentration of $NO_x$ in the atmosphere decreased instantaneously upon the
cessation of vehicular traffic. Meanwhile, although the daily mean concentration of $O_3$ did not change significantly
after January 23rd (Fig. 2b), the daily range of variation in its concentration (i.e. the difference between the
minimum and maximum $O_3$ concentrations in a day) did decrease significantly after this date.

The variations in daily mean temperature, daily mean RH, daily mean wind speed, and daily dominant wind

direction during the study period are shown in Figs. 2d and 2e. As evident in Fig. 2d, the RH and temperature
correlated strongly with each other, indicating that the dry air in the PRD comes predominantly from cold air
masses. During the study period, cold fronts occurred on 26–27 December 2019, 12 January 2020, and 27–30
January 2020. Whenever a cold air mass passes over the Shiyan Base, the daily mean temperature and RH will
decrease in step with each other. As evident in Fig. 2e, the daily mean wind speeds of the study period were usually
below 2 m/s. Additionally, the daily dominant wind direction was in the northerly direction for approximately 75%
of the time. The weather that was observed during the study period is common during the winters in Shenzhen,
indicating that no meteorological abnormalities had coincided with the study period. When we compared the
variations in each meteorological factor against the pollutant concentrations, only the RH and $O_3$ concentration
were found to be significantly (negatively) correlated with each other. The daily mean concentrations of $PM_{2.5}$ and



NO$_x$ were not significantly correlated with any meteorological factor during the study period.

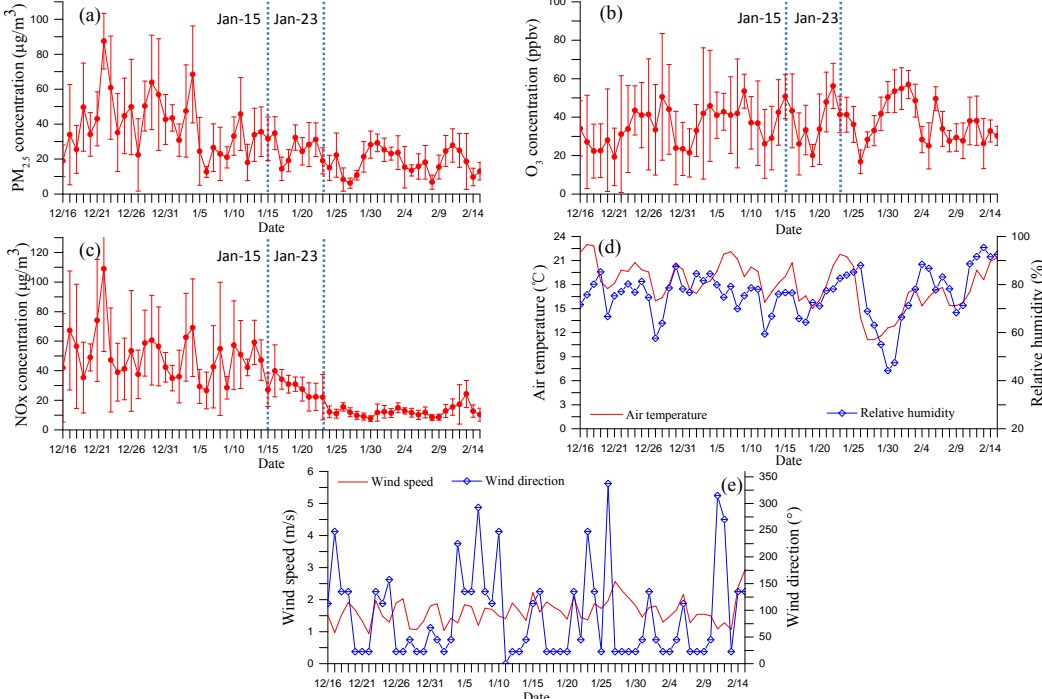


**Fig. 2.** Daily variations in pollutant concentrations and related meteorological factors in the surface layer during the
period of 16 December 2019 to 15 February 2020: (a) PM$_{2.5}$ concentrations, (b) O$_3$ concentrations, and (c) NO$_x$
concentrations observed at different heights; (d) Air temperature and relative humidity observed at Shiyan
Meteorological Observation Base; (e) Wind speed and wind direction observed by the auto weather station at
Shiyan Meteorological Observation Base. ppbv means parts per billion by volume.

Taking 23 January 2020 as a date boundary, Fig. 3 compares the average concentrations of the 3 pollutants

before and during the lockdown. Fig. 3 clearly illustrates the decrease of PM$_{2.5}$ and NO$_x$, during the lockdown. The
decrease of NO$_x$ is much more drastic than that of PM$_{2.5}$. The change of O$_3$ is more complex than that of PM$_{2.5}$ or
$NO_x$. The daily average $O_3$ concentration had slightly increased during the lockdown, which is consistent with the
findings of other studies (Gualtieri et al., 2020). While the mean values of the daily maximum 8 h average $O_3$
($MDA8O_3$) had significantly decreased during the lockdown. The definition of $MDA8O_3$ is as follows: in a natural
day, take 0:00, 1:00,..., 16:00 local standard time (LST) as the starting point respectively, calculate the average
concentration of $O_3$ for 8 consecutive hours for each starting point, and one can obtain totally 17 8-hour-average $O_3$
concentrations. The maximum value of all the 8-hour-average concentrations is $MDA8O_3$, which is generally used
to assess the severity of $O_3$ pollution. The truth that daily $O_3$ concentration and $MDA8O_3$ had different changes
means there might be quite different chemical environments related to $O_3$ before and during the lockdown.

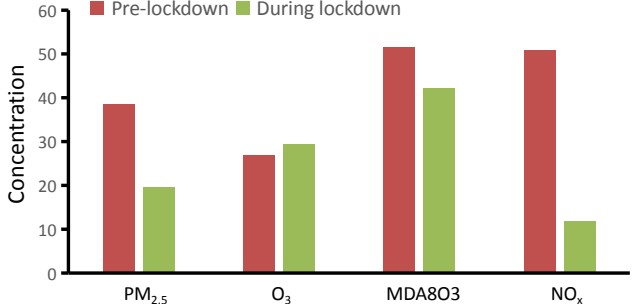


**Fig. 3.** Comparison of the average values of the $PM_{2.5}$ (in µg/m³), $O_3$ (in ppbv), MDA8O3 (in ppbv) and $NO_x$ (in
µg/m³) concentrations of the whole surface layer before and during the lockdown

Table 1 furtherly compares the average values of the meteorological factors and pollutant concentrations
during and before the lockdown and those in the December of 2017 (Li et al., 2020). In the pre-lockdown period,
the air quality in the area where the SZMGT is located had already been significantly improved compared with
December 2017, which is reflected in the decrease of the average concentrations of the three pollutants.
Table 1 also provides the information on the changes of meteorological factors. In December 2017, the relative

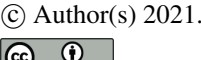



humidity was lower than that in the period of the current study, which was more favorable to the photochemical
reactions generating PM$_{2.5}$ and O$_3$. On the other hand, the wind speed in December 2017 was much higher than that
in the period of the current study, which was favorable to disperse the pollutants. Thus, it is difficult to compare the
comprehensive impacts of the meteorological conditions on the pollutant's concentrations in December 2017 with
those in pre-lockdown or during-lockdown period.

**Table 1.** Comparison of pollutants and meteorological elements during lockdown, before lockdown and December

2017

| Time period | December 2017 | Before 23 Jan. 2020 | After 23 Jan. 2020 | Change of pre-lockdown compared with December 2017 | Change during lockdown compared with before lockdown |
|---|---|---|---|---|---|
| PM$_{2.5}$ (µg/m$^3$) | 47.0 | 38.5 | 19.5 | -18.1% | -49.4% |
| O$_3$ (ppbv) | 42.0 | 26.8 | 29.4 | -36.2% | +9.7% |
| MDA8O$_3$(ppbv) | 59.6 | 51.4 | 42.1 | -13.8% | -18.6% |
| NO$_x$ (µg/m$^3$) | 54.2 | 50.9 | 11.8 | -6.1% | -76.8% |
| Air temperature (℃) | 17.1 | 18.9 | 16.5 | 10.5% | -12.7% |
| Relative humidity (%) | 58.5 | 75.3 | 77.0 | +28.7% | +2.3% |
| Wind speed (m/s) | 2.2 | 1.6 | 1.8 | -27.3% | +12.5% |
| Wind direction | NNE | NNE | NNE | — | — |

* NNE means north-north-east. All pollutant concentrations in the table are average values for the whole surface
layer recorded by the tower.

While, at least one of the possible reasons leading to the decrease of pollutants concentrations in the
pre-lockdown period compared with December 2017 is quite clear, which is the strengthening of pollution emission
control in Shenzhen in the past 2 years. For example, according to local news reports, in 2019 more than 1000
heavy-duty diesel trucks in Shenzhen were replaced by electric trucks. In the past, the emissions of these diesel





trucks were an important source of pollution in Shenzhen.

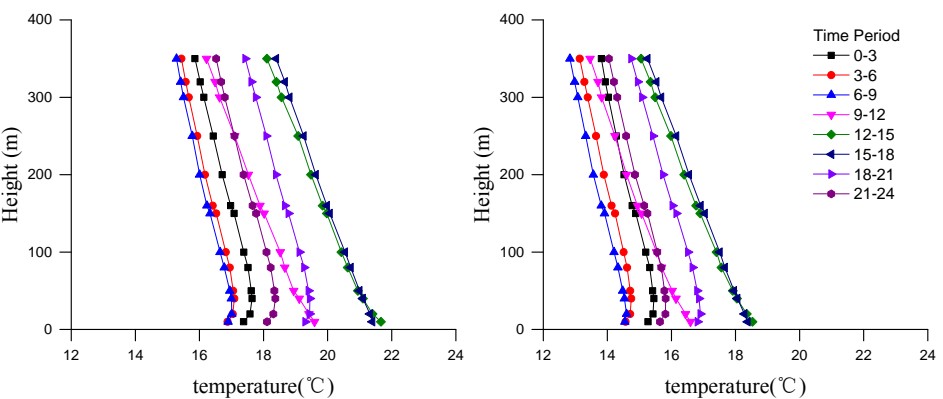


**Fig. 4.** Average vertical air temperature profiles recorded by the Shenzhen Meteorological Gradient Tower before
and after 23 January 2020.

The changes in the meteorological factors during the lockdown compared with pre-lockdown period may

largely be attributed to the intense cold air front that developed on 27–30 January 2020, which decreased the mean
temperature of the lockdown period. By contrast, the RH changed very little after the lockdown was implemented.
The meteorological factors that were most closely related to pollutant dispersal in the Shenzhen region were the
wind speed and wind direction. Although the average wind speed increased during the lockdown, it was still weaker
than that in December 2017 and never exceeded 2.0 m/s, thereby limiting any improvement in pollutant dispersion.
The dominant wind direction in the pre-lockdown and lockdown periods was north-north-east, indicating that the
winds in Shenzhen came mainly from the inland regions of China. In a normal year, these winds would carry a
large amount of air pollution from the inland parts of the PRD and thus cause a spike in pollutant concentrations (Li
et al., 2020). Therefore, it can be concluded that the meteorological conditions of the Shenzhen region were largely
identical before and during the lockdown. Although an intense cold spell occurred after January 23rd and the



average wind speed had increased slightly, it can be learnt from the experience that these changes would not be
nearly enough to cause the dramatic decreases in average $PM_{2.5}$ and $NO_x$ concentrations recorded. Figure 4
furtherly provides the average vertical air temperature profiles recorded by SZMGT before and after 23 January
2020, from which it can be found that there is no significant difference in the stratification of air temperature before
and after the outbreak of the pandemic. The data illustrated in Table 1 and Fig. 4 show that the drastic change of the
pollutant concentration in the study period is almost impossible to be caused by the change of meteorological
factors.
**3.2 Diurnal Variations at Different Heights**
Figure 5 shows the diurnal variations in $PM_{2.5}$, $NO_x$, and $O_3$ concentrations on the surface (2 m) and at three
different heights of the SZMGT (120, 220, and 335 m) before and during the lockdown. The $PM_{2.5}$ time series
curves in Fig. 5a are characterised by two trends: a bimodal distribution for the ground level (2 m) and 120 m
curves, and a unimodal distribution for the 220 m and 335 m curves. The peaks of the bimodal curves occurred at
09:00LST and 20:00 LST, which correspond roughly to the morning and evening rush hours. The difference
between the $PM_{2.5}$ curves at 0 m/120 m and that of 220 m/335 m probably reflects the uplift process of the mixing
layer top in this area. During night and early morning, the height of the mixing layer top is between 120m and
220m, so the curves of the upper and lower layers are quite different. After the noon time, with the rise of the
mixing layer top, the curves of all layers become to be similar. Although the $PM_{2.5}$ concentrations at 2 and 120 m
followed the same qualitative trend, the values on the ground were generally lower than those at 120 m. This may
have been caused by the presence of dense forests near the ground observation point (Shiyan Base), which may
have obstructed the dispersal of particulate matter and thus reduced the apparent $PM_{2.5}$ concentration. The peaks of
the unimodal 220 and 335 m curves occurred at 17:00–19:00 LST. Therefore, the diurnal variations in $PM_{2.5}$
concentration were different at lower and higher heights. This is consistent with the findings of Li et al. (2020),





whose study implied that high- and low-height $PM_{2.5}$ may have different sources. High-height $PM_{2.5}$ is formed
predominantly by chemical reactions, whereas low-height $PM_{2.5}$ may be derived from multiple sources
(predominantly surface-level primary emissions).

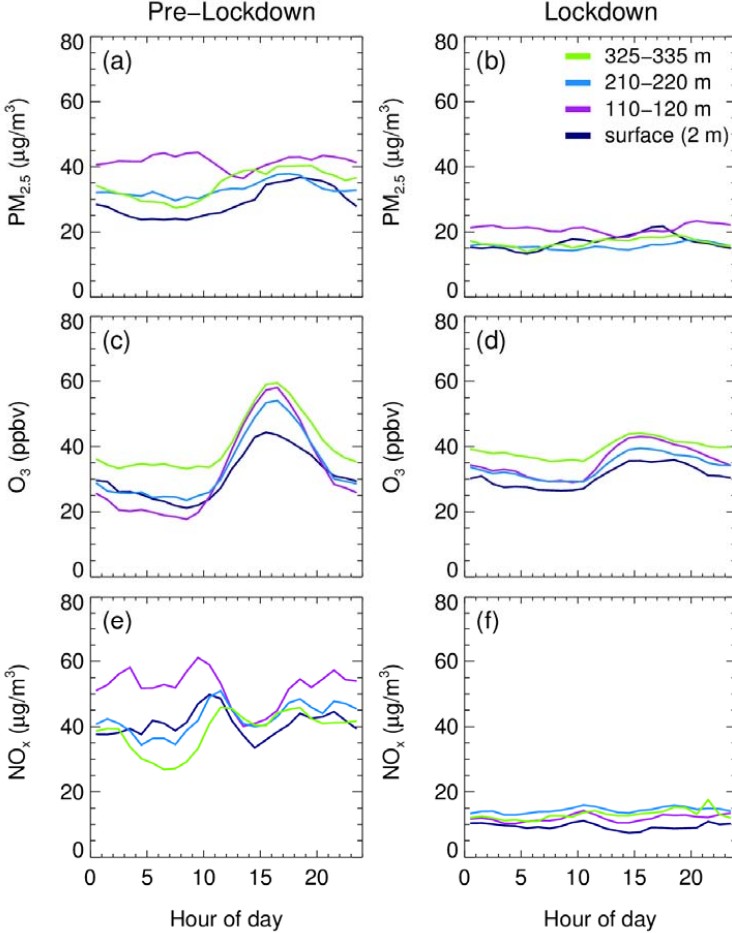


**Fig. 5.** Diurnal variations in the pollutants observed at different heights of the meteorological tower: $PM_{2.5}$

concentrations before lockdown (a) and during lockdown (b); $O_3$ concentrations before lockdown (c) and during
lockdown (d); $NO_x$ concentrations before lockdown (e) and during lockdown (f).





The diurnal variations in $PM_{2.5}$ concentration at 2, 120, 220, and 335 m during the COVID-19 lockdown are
shown in Fig. 5b. It was obvious that the $PM_{2.5}$ concentration had decreased significantly at all heights after
January 23rd. The 120 m curve still had the highest $PM_{2.5}$ concentration and it still retained the bimodal structure of
its pre-lockdown counterpart. The $PM_{2.5}$ concentrations at 220 and 335 m were still unimodal, and the peak still
occurred at a similar time. The biggest lockdown-mediated change in $PM_{2.5}$ concentration occurred at 2 m, where
the curve lost its peak in the morning and changed from a bimodal to a unimodal graph. It is likely that the morning
peak of the pre-lockdown curve was caused by direct emissions from nearby human activities. These emissions
were therefore greatly reduced by the lockdown-mediated decrease in human activity and were more easily blocked
by the dense forest around the ground observation point.
With regard to $O_3$, it was evident that the diurnal variation in its concentration was unimodal and peaked at
approximately 15:00–16:00 LST (when photochemical $O_3$ formation is most active) both before and during the
lockdown (Figs. 5c and 5d, respectively). These diurnal variations were also qualitatively invariant with altitude;
that is, only the average concentration varied from one altitude to the other. However, the shape of the $O_3$ curve did
become significantly flatter during the lockdown, indicating that the range of the diurnal variations became much
narrower during the lockdown. The flattening of the peaks and valleys of the $O_3$ curve implies that the chemical
reactions that generate $O_3$ during day-time and consume $O_3$ during night-time became to be mush inactive during
the lockdown. Under this condition, the $O_3$ concentration seemed to be determined primarily by background $O_3$
concentration (Xu et al., 2020). While, it should be noted that a flatter $O_3$ curve means the decrease of $MDA8O_3$,
which implies that the prevention of $O_3$ and $PM_{2.5}$ pollution can be realized at the same time theoretically.
In the case of $NO_x$, the diurnal variations in its concentration were bimodal before the lockdown (Fig. 5e). The
2 and 120 m curves showed a peak at 09:00 LST, which coincided with the timing of the morning rush hour. The
second peak, which begins at 18:00 LST and continues until 21:00 LST, was likely caused by the evening rush hour



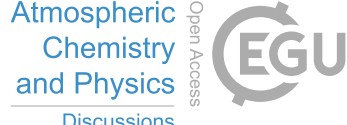

and the night-time decrease in the altitude of the mixed layer. The first peak in both the 220 and 335 m curves
lagged the first peak of the curves of lower altitude by 1 hour. However, the second peak occurred at roughly the
same time in both sets of curves. Although the mean $NO_x$ concentrations had decreased significantly during the
lockdown, their diurnal variations were still bimodal (Fig. 5f). However, inter-altitude differences in $NO_x$
concentration did become much smaller during the lockdown and the timing of the $NO_x$ peaks at each altitude also
became much closer to each other. During the lockdown, the first peak was delayed by 1 hour while the second
peak occurred at 17:00–19:00 LST. The $NO_x$ concentrations also changed in another significant way; that is, they
were lower at 2 and 120 m than at 220 and 335 m. Since $NO_x$ is a primary pollutant, its significantly lower
concentrations at the low altitudes implies that near-ground chemical reactions consume it more rapidly than the
high-altitude chemical reactions do.

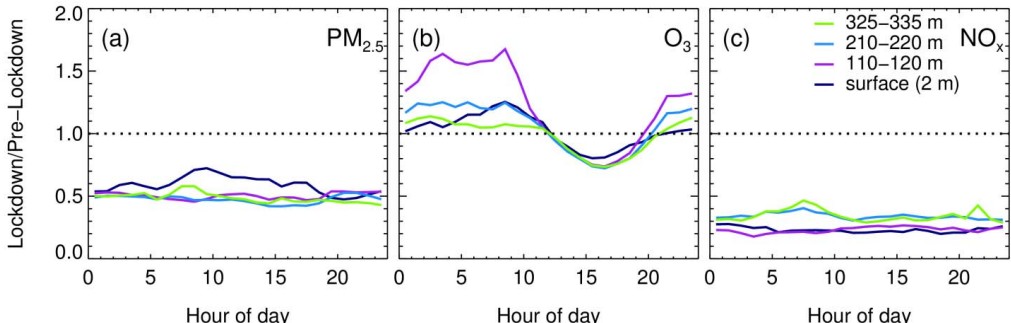


**Fig. 6.** Diurnal variations in the during-lockdown/pre-lockdown ratios of the pollutants observed at different

heights of the meteorological tower: (a) $PM_{2.5}$; (b) $O_3$ and (c) $NO_x$.

In order to furtherly analyze the change of the pollutants during the lockdown, the concentration ratios of the
pollutants before and during the lockdown are calculated, and the diurnal variation of the ratios is illustrated in Fig.
6. The diurnal variation curves of different pollutants had different characteristics. For $NO_x$, the curves at different



heights were relatively consistent, which were relatively flat and maintained around 0.3, indicating that $NO_x$
decreased significantly and evenly in the boundary layer. The curves for $PM_{2.5}$ were different. The ratio curves were
relatively flat and maintained at about 0.5 all day at heights above 110 m, but there were relatively large
fluctuations on the ground. The ratio on the ground increased significantly between 7:00 and 18:00 LST instead of
keeping flat. Especially during 8:00 to 10:00 LST in the morning, the ratio value reached around 0.7, which showed
that the decrease of ground level $PM_{2.5}$ concentration ($\sim$ -30%) during the morning "rush-hours" of lockdown was
not as drastic as that of the average data of the whole boundary layer ($\sim$ -50%), though it was still difficult for the
$PM_{2.5}$ generated on the ground to affect the air mass above 100 m. The fluctuation of ratio diurnal curves of $O_3$ was
much more obvious than those of the other two pollutants. The ratios were generally greater than 1.0 in night and
less than 1.0 in daytime, which showed that during lockdown, the concentration of $O_3$ increased in night and
decreased in daytime. Especially at the height of 110-120 m, the fluctuation of the curve was more drastic, and the
maximum ratio in night could reach 1.7. A possible reason leading to this phenomenon is that in the area where the
SZMGT is located, the key height of night chemical reactions may be around 110-120 m. Under normal conditions,
the night chemical reactions consuming $O_3$ in this layer may be more active than other heights. Therefore, when all
emissions were weakened, the $O_3$ consumed by night chemical reaction was greatly reduced, and the $O_3$
concentration in this layer increases significantly. While this conjecture need further researches to confirm in the
future.
**3.3 Vertical Distribution of Pollutants**
The changes in the vertical distribution of the 3 pollutants and total oxidants, Ox (= $O_3$ + $NO_2$), measured at
120, 220, and 335 m of the SZMGT before and during the lockdown are shown in Fig. 7. In terms of the all-day
averages (Figs. 7a–7d), it was obvious that the $PM_{2.5}$, $NO_x$ concentrations were lower at all altitudes during the
lockdown. By contrast, the $O_3$ concentrations did not decrease significantly, but their vertical gradations did



become less pronounced. Therefore, the $O_3$ concentrations became more uniform in the vertical direction during the
lockdown. The Ox concentrations, both on daytime and nighttime average (Figs. 7h–7i), also were generally lower
during the lockdown than that before the lockdown, indicating that the oxidation capacity for the whole boundary
layer weakened during the lockdown. We also checked the nitrate radical production rate during the nighttime,
which is an indicator of nighttime oxidation reactions, and showed a large decline with an average of ~70%,
suggests the weaken $NO_3$ oxidation capacity. The decrease in nighttime oxidation is mainly attributed to cliff fall of
$NO_x$. Overall, our vertical observation showed that the atmospheric oxidation processes, including photochemistry
and nighttime chemistry were largely reduced due to the lockdown.

As shown in Fig. 7a, the $PM_{2.5}$ concentrations initially decreased with increasing altitude, before increasing

slightly with further increases in the altitude. This trend occurred both before and during the lockdown period. The
$PM_{2.5}$ concentration was the highest at the lowest observation point studied (i.e. 120 m), whereas the concentration
at 335 m was between those recorded at 120 and 220 m. This observation is rather interesting, as it is contrary to
the expectation that the $PM_{2.5}$ concentration should decrease monotonically with increasing altitude (Sun et al.
2010). However, this can be explained if we consider the results of previous studies about the possible sources of
$PM_{2.5}$ at each altitude. At the lowest height (120 m), $PM_{2.5}$ may have come from photochemical reactions and
primary pollution sources on the ground. At the middle level and above, $PM_{2.5}$ is formed mainly by photochemical
reactions (Li et al. 2020). Therefore, the efficiency of $PM_{2.5}$ generation at these heights may be affected by the
oxidative potential of the atmosphere. Based on the observations on the SZMGT, the $O_3$ concentration generally
increases with increasing height, and the Ox is also higher at 335 m than at 220 m. Hence, it is likely that the
oxidation capacity of the atmosphere is higher at the highest level, increasing the efficiency of $PM_{2.5}$ formation at
this altitude. Although volatile organic compound (VOC) concentrations were not measured on the SZMGT,
measurements in the nearby region of Taiwan have shown that these compounds also tend to increase with





increasing altitude, up to a peak of 300–400 m, thus providing an ample supply of reactants for photochemical
reactions at high altitudes (Vo et al., 2019). At the middle level (220 m), the $PM_{2.5}$ concentration is not significantly
affected by primary pollutant sources on the ground, and the $PM_{2.5}$-forming photochemical reactions are also less
efficient here than in the higher levels. Consequently, the $PM_{2.5}$ concentrations are lower in the middle level than in
the high level.

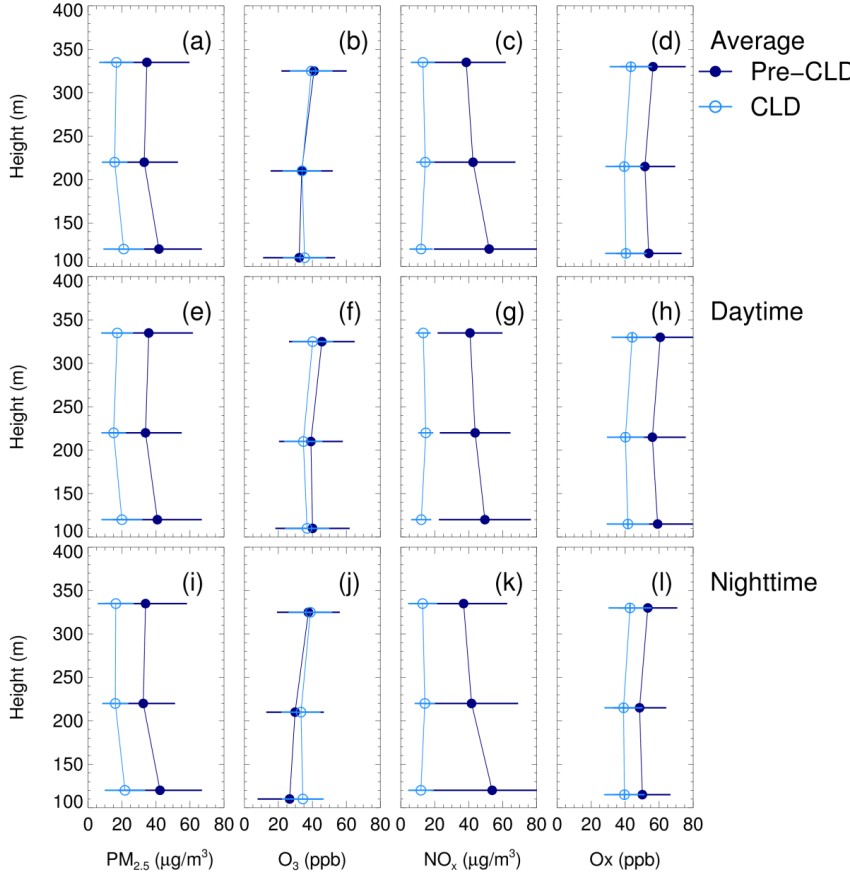


**Fig. 7.** Vertical distribution of three pollutants and Ox (= $NO_2$ + $O_3$) observed at the Shenzhen Meteorological
Gradient Tower. Panel (a-d) show whole-day data; panel (d-h) show day-time data; and panel (i-l) show night-time
data.
As mentioned above, the $O_3$ concentrations increased monotonically with increasing altitude (Fig. 7b). Even
during the lockdown, the average concentration of $O_3$ stayed high without showing any significant change.
Completely different from the change of $O_3$, vehicular exhaust-gas emissions had plummeted to a very low level
during the lockdown, which is clearly evidenced in Fig. 7c. This figure shows that the $NO_x$ concentrations had
decreased considerably at near-ground altitudes, especially at 120 m, where the concentration had decreased by
over 75% compared with pre-lockdown levels. The persistence of high $O_3$ concentrations during the lockdown
period proves the importance of VOCs for the air quality of this region once again. Actually, given the significant
decrease in $NO_x$ concentrations (as much as -78.2% in the current study), the concentrations of VOCs did not seem
to have a same change as $NO_x$ did. In recent studies, Qi et al (2021) reported that the decrease of VOCs in PRD
during the lockdown is much less than that of $NO_x$, and Liu et al (2021) reported that formaldehyde (HCHO)
abundance in the PRD area even slightly increased during the lockdown based on TROPOspheric Monitoring
Instrument (TROPOMI) satellite observation, which indicate that VOCs were likely to play an even more important
role in photochemical reactions during the lockdown period.
Figs. 7e–h and 7i–l displays the vertical distributions of pollutants and Ox during the day and night,
respectively. The day-time and night-time distributions of $PM_{2.5}$, $NO_x$ and Ox were not significantly different. By
contrast, the vertical distribution of $O_3$ varied significantly between day and night. At all altitudes, the day-time $O_3$
concentrations were generally lower during the lockdown, whereas the night-time concentrations were higher. This
paradoxical trend can be explained by the weakening of atmospheric chemical activity during the lockdown. Since
the decrease in human activity during the lockdown also decreased primary pollutant emissions, the availability of
precursors for photochemical $O_3$ generation was significantly lower, resulting in decreased day-time $O_3$
concentrations. During the night, the dark chemical reactions that consume $O_3$ also became less active during the
lockdown, which resulted in significantly higher night-time $O_3$ concentrations at the near-surface atmosphere.





These changes are consistent with the diurnal variations in $O_3$ (Fig. 5d).

### 3.4 Correlations at Different Altitudes

Fig. 8 depicts scatter plots and fit lines of $O_3$ versus $PM_{2.5}$ at each level of the SZMGT, before and during the
lockdown.

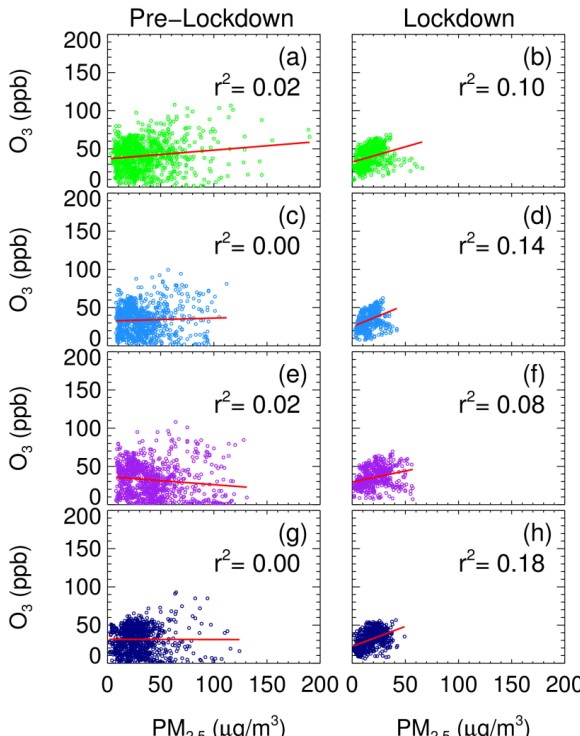


**Fig. 8.** Scatter plots of the $PM_{2.5}$ and $O_3$ concentrations at different heights of the meteorological tower before and

during the lockdown: (a) and (b) are for ground level; (c) and (d) are for low level; (e) and (f) are for middle level;
(g) and (h) are for high level. The fit lines of the plots were produced following the instruction of Cantrell (2008).

Prior to the lockdown, the correlation between the $O_3$ and $PM_{2.5}$ concentrations were weak, with none of the
correlation coefficients (*R*) passing the significance test. When performing the correlation significance test for two



variables, namely x and y, which means two different pollutants, the Pearson correlation coefficient $R$ was
calculated. Compare $R$ to the appropriate critical value corresponding to the N-2 value in a standard table, where N
is size of the sample set of (x, y) pairs. If the absolute value of R is greater than the critical value, the correlation
between the two variables is significant. During the lockdown, the correlation between $PM_{2.5}$ and $O_3$ became
significantly stronger, with the R values for the 0, 120, 220, and 335 m scatter plots all being significant at the 0.1
level. It may be inferred that prior to the lockdown, $PM_{2.5}$ and $O_3$ did not have related sources. However, during the
lockdown, both were likely to have a similar source. In the PRD region, VOCs contribute significantly to the
formation of fine particles (Liu et al., 2008; Zheng et al., 2009), especially secondary organic aerosols (Huang et al.,
2006; Chang et al., 2019; Zhang et al., 2019). Although the advent of the COVID-19 pandemic did result in
reductions in the concentrations of $NO_x$, $SO_2$, and other primary pollutants, the VOCs emissions might not change
as dramatically as $NO_x$ (Liu et al., 2021), which provided important precursors for both $O_3$ and the secondary
organic aerosols during the lockdown, and strengthened the correlation between the $PM_{2.5}$ and $O_3$ concentrations.

Li et al. (2020) analysed the correlation coefficients of $O_3$ and $PM_{2.5}$ at different heights of SZMGT in

December 2017, and the conclusions were different from this study. In December 2017, the correlation coefficient
of $O_3$ and $PM_{2.5}$ increased significantly with the increase of height. They pointed out that this is because $PM_{2.5}$ is
also mainly generated by photochemical reaction at high altitudes, so it has a strong correlation with $O_3$. At lower
heights, a considerable part of $PM_{2.5}$ is primary source and had nothing to do with photochemical reactions, so it
had much weaker correlation with $O_3$. While in this study, the correlation between $PM_{2.5}$ and $O_3$ was weak at all
heights in the pre-lockdown period. The reason leading to this is mainly because that Shenzhen had conducted a
large number of pollution emission control strategies in the past two years, resulting in a significant decrease in the
primary pollutants. As shown in Table 1, the average concentration of $PM_{2.5}$ in the whole surface layer during the
pre-lockdown period had decreased by 18.1% compared with December 2017, while the average concentration of



O$_3$ had decreased by 36.2%. The decrease of O$_3$ concentration in the surface layer was twice that of PM$_{2.5}$, while
the primary aerosol like black carbon is not reduced, indicating that there were much fewer products of
photochemical reaction and secondary aerosol formation, so that the correlation coefficient between O$_3$ and PM$_{2.5}$
concentration was no longer high even in the higher heights. However, during the lockdown, the average
concentration of PM$_{2.5}$ decreased again because the primary emission was drastically compressed. In this process,
PM$_{2.5}$ formed from the primary emission became to be insignificant, and the photochemical oxidation of VOCs
became to be important sources of particulate matter again. Therefore, the correlation coefficients between O$_3$ and
PM$_{2.5}$ became to be higher than those before the lockdown.

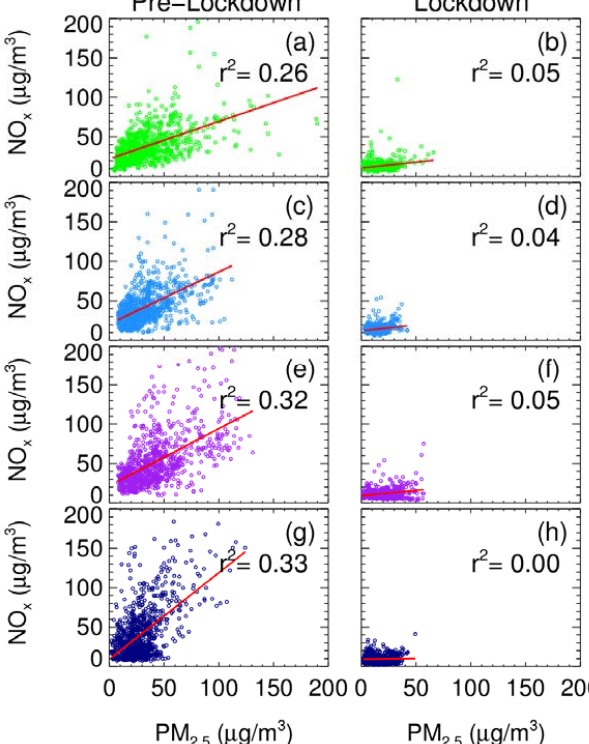


**Fig. 9.** Scatter plots of the PM$_{2.5}$ and NO$_x$ concentrations at different heights of the meteorological tower before ad
during the lockdown: (a) and (b) are for ground level; (c) and (d) are low level; (e) and (f) are for middle level; (g)





and (h) are for high level.

Fig. 9 compares the correlation between the $PM_{2.5}$ and $NO_x$ concentrations before and during the lockdown.

The trend of the correlation between $PM_{2.5}$ and $O_3$ was the exact opposite of that between $PM_{2.5}$ and $O_3$; that is, it
was strong before the lockdown (R = ~0.5 at all altitudes) but much weaker after (R = ~0.2 at all altitudes).
Therefore, there may be significant differences between $PM_{2.5}$ sources before and during the lockdown. Owing to a
lack of data, it was not possible to perform a composition analysis to determine why $PM_{2.5}$ was closely correlated
with $NO_x$ emissions prior to the lockdown but not during it. One possible explanation is that the primary emission
of $PM_{2.5}$ in the PRD is a large part before the lockdown, since NOx can be treated as an indicator of anthropogenic
emission, and primary emission of $PM_{2.5}$ decreased significantly during the lockdown. The other possible
explanation is that the nitrate content of $PM_{2.5}$ in the PRD decreased significantly during the lockdown, given that it
has been previously found that nitrate accounts for a large percentage of $PM_{2.5}$ before the year of 2020 (Yang et al.,

2020).

Fig. 10 displays the correlation between the $O_3$ and $NO_x$ concentrations before and during the lockdown. Prior

to the pandemic, $O_3$ and $NO_x$ were negatively correlated with each other owing to $NO_x$ titration. The relationship
between the $O_3$ and $NO_x$ concentrations could be fitted with an exponential function. During the lockdown, the
(negative) correlation between $O_3$ and $NO_x$ weakened significantly, which means that at very low $NO_x$
concentrations, variations in the concentration of this pollutant seemed to virtually have no obvious effect on the $O_3$
concentrations.

The comparison of scatter plots before and during the lockdown showed that $PM_{2.5}$ was poorly correlated to $O_3$

but closely correlated to $NO_x$ before the lockdown, indicating that a large proportion of $PM_{2.5}$ might come from
primary emissions or nitrate aerosol. While after the implementation of the lockdown, $PM_{2.5}$ became to be closely
correlated to $O_3$, but not to $NO_x$, and $O_3$ formation cannot attribute to the local photochemistry, indicating that
PM$_{2.5}$ during the lockdown might primarily be secondary pollutants (especially organic aerosols) generated from
photochemical reactions.

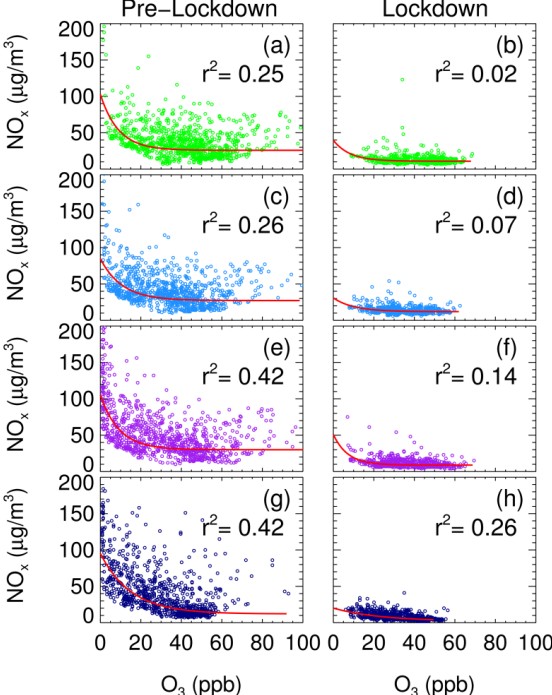


**Fig. 10.** Scatter plots of the O$_3$ and NO$_x$ concentrations at different heights of the meteorological tower before and
during the lockdown: (a) and (b) are for ground level; (c) and (d) are r low level; (e) and (f) re for middle level; (g)
and (h) are for high level.

**4 Conclusions and Implications**

In this study, changes in the NO$_x$, O$_3$, and PM$_{2.5}$ concentrations over the PRD, mediated by the local

COVID-19 pandemic lockdown measures, were investigated through the analysis of their vertical distribution
before and during the lockdown, using data from the Shiyan Base and SZMGT. The conclusions of this study are as
follows:





(1) The advent of the COVID-19 pandemic forced a dramatic decrease in human activity. This greatly reduced
the emission of primary pollutants like $NO_x$, thus changing the chemical environment of the near-surface
atmosphere. The concentration of $PM_{2.5}$ was also reduced significantly because of the decrease in precursor
availability.
(2) The reduction in primary pollutant emissions during the COVID-19 pandemic lockdown significantly
decreased $MDA8O_3$, while did not decrease the daily average concentration of $O_3$. The diurnal curves of $O_3$
concentration were changed by the lockdown, with the day-time concentrations being lower and the night-time
ones being higher than the pre-pandemic concentrations at all levels of the SZMGT.
(3) The correlation between $PM_{2.5}$ and $O_3$ concentrations was insignificant before the lockdown but became
significantly stronger after, to the point where the correlation coefficients between $PM_{2.5}$ and $O_3$ were significant at
the 0.05 level, regardless of altitude. This indicates a strong correlation between $PM_{2.5}$ and $O_3$. By contrast, the
correlation between $PM_{2.5}$ and $NO_x$ was much weaker during the lockdown. Hence, the composition of $PM_{2.5}$ may
have changed from being predominantly from primary emissions or nitrate aerosol before the lockdown to being
predominantly a secondary organic aerosol thereafter. However, the validation of this hypothesis will require
further investigations.
(4) Prior to the COVID-19 pandemic, the $O_3$ and $NO_x$ concentrations were significantly negatively correlated
with each other. This correlation virtually disappeared after the beginning of the pandemic. It may be concluded
that at very low $NO_x$ concentrations, variations in its concentration have almost no effect on the $O_3$ concentration.
Overall, the advent of COVID-19 has devastated economies and societies around the world. However, the
dramatic reduction in human activity resulting from the lockdown measures has provided a unique opportunity for
researchers to study the response of the atmospheric environment to human activities. The data indicate that the
atmospheric chemical environment of the PRD has changed during the pandemic, leading the drastic change of

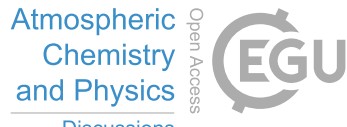

pollutants concentrations. These results have a clear indication for pollution prevention policy. In the past, quite a
few environmental policy studies doubted whether it was necessary to further reduce mobile emissions, because in
some areas, decreasing $NO_x$ led to an increase of $O_3$ concentration. While this study shows that the continuous
reduction of $NO_x$ emission can still reduce the peak value and MDA8$O_3$, although it will not further reduce the
daily average value of $O_3$.

**Competing interests.**
The authors declare that they have no conflict of interest.

**Acknowledgement.**
This study is supported by the Science and Technology Projects of Guangdong Province (grant number
2019B121201002), Natural Science Foundation of China (grant number 42075059, 41907185) and Guangdong
Basicand Applied Basic Research Foundation (grant number 2019A1515012008)

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
