# Peer review of "Impact of the COVID-19 pandemic on the observed vertical distributions of"

_Atmospheric Chemistry and Physics, 2021_

## Author Comment (AC1)

**Reviewer #1**

The outbreak of the 2019 novel coronavirus (COVID-19) has brought tremendous impact on human health and social economy. Sharp declines in primary pollution provided a unique chance to examine the relationships between anthropogenic emissions and air quality. The author investigated the vertical structure of pollutants by the highest meteorological tower in ShenZhen City. They found that $O_3$ concentrations were not sensitive to $NO_x$ concentrations during lockdown, which implies that $O_3$ levels during the lockdown are more representative of the regional background. They deduced that reductions of anthropogenic emissions are effective to decline $PM_{2.5}$ and $O_3$ pollutant levels in the Pearl River Delta. Minor revisions are required before acceptance. Comments:

Thanks for your positive comments. The responses to your comments are listed as below:

1. How are the instruments on the tower calibrated and maintained on the meteorological tower to ensure data quality. The methods need to be explained in the second section?

The instruments on the tower are maintained by professional service providers and are routinely exchanged for calibration once every 3 months.

2. In Figure 5, why are the concentrations of $PM_{2.5}$, $O_3$ and $NO_x$ higher up than at the surface?

The pollutants concentrations on the ground are from an atmospheric environmental observation station lying at the bottom of the SZMGT. The height of the sampling port of the ground station is lower than that of the surrounding forest top, which can absorb the pollutants transported from the sources beyond the forest and lead to lower concentration values than on the tower.

3. In Figure 9 and 10, how about the local photochemistry in different periods?

Fig. 9 compares the correlation between the $PM_{2.5}$ and $NO_x$ concentrations before and during the lockdown, while Fig. 10 displays the correlation between the $O_3$ and $NO_x$ concentrations before and during the lockdown. During the pre-lockdown period, the photochemical reaction generating the $PM_{2.5}$ must be very active, and the $NO_x$ from traffic provided rich precursors for the photochemical reaction. Some recent studies pointed out that $NO_3$- is an important component of water-soluble aerosols in this region (Wu et al., 2020; Yang et al., 2021), which is quite accordant to the results shown in Fig. 9. At the same time the titration effect was also very obvious during the pre-lockdown period. While during the lockdown period, the photochemical reactions generating $PM_{2.5}$ were quite inactive, and the titration effect was no more obvious since the emission of $NO_x$ had been reduced greatly.

**References:**

Yang, H., Zhang, Y., Li, L., et al. Characteristics of aerosol pollution under different visibility conditions in winter in a coastal mega-city in China. Journal Tropical Meteorology,, 26(2): 231-238. 2020.

Wu,L., Wang,Y., Li,L., Zhang,G.. Acidity and inorganic ion formation in $PM_{2.5}$ based on continuous online observations in a South China megacity. Atmospheric Pollution Research, 11: 1339-1350.2020.

4. It is suggested to add motor vehicle data in the article, to explain the change of emission from pre-lockdown to lockdown.

Though for the limitation of the data source, it is difficult to get exactly accurate vehicle data, the traffic data can still be estimated in light of the public new report. A nationally popular navigation service provider announced that during lockdown period, the traffic flow was about 14.1% of the normal situation,

which means there were around 282,000 vehicles running in the whole city of Shenzhen with territory of 2000 km$^2$, while the normal average number is around 2000,000 vehicles everyday.

---

## Author Comment (AC2)

**Reviewer #2**

**General Comments**

The Covid-19 lockdown provides a unique opportunity for assessing the effects of substantial emission reductions on atmospheric chemistry. This paper used ambient measurements from a tower situated in the Pearl River Delta of China to explore the response of air pollutants to the lockdown. While this paper is within the scope of ACP, the present paper is limited to a cursory data analysis, without significant contribution to our existing knowledge. The absence of sound analysis accompanied by a lack of in-depth discussion of the observed phenomenon make this paper unpublishable in the present form. Besides the lack of novel insights, I found the manuscript overall hard to follow due to lots of typos throughout the manuscript. While addressing the specific comments below may improve the paper, I don't think these improvements could justify publication in ACP. Therefore, I would recommend this paper to be rejected. Thanks for your comments. We have increased the depth of analysis, and collect and cite as many studies on atmospheric environment in PRD area as possible to support our conclusions in the revised manuscript. We hope that these efforts will make the manuscript more publishable.

**Major comments:**

1) Line 33-34: Why photochemical reactions are not considered as a significant ozone source? While anthropogenic emissions are low during the lockdown period, the oxidation of biogenic VOC still contributes to the ozone formation given the elevated BVOC emissions over the PRD.

The fact supporting that photochemical reactions are not the main source of ozone is that the diurnal curve is much flatter during the lockdown period than during the pre-lockdown period, and the MDA8O3 values during the lockdown period is much lower than that of the pre-lockdown period, which means the chemical reactions generating and consuming  $O_3$  are much more inactive during the lockdown period. Furthermore, in winter, the BVOC emission from plant is much weaker than in summer. However, we will revise the statement of Line 33-34 and will emphasize possible effect of BVOC on generating  $O_3$ , especially on generating the weak peak of  $O_3$  diurnal curve at noon.

2) Line 150-158: Please consider simplifying these statements since they are not key contents regarding scientific publication.

The reason why the whole process of lockdown measures is introduced in detail is to help understand the reasons leading to the variation of pollutant curves in Figure 2. We have simplified it as follow.

"The first case of COVID-19 in Shenzhen was reported by local news outlets on 15 January 2020, which is the starting date for the reduction of citizens' outdoor activities. On 23 January, the Guangdong province, where Shenzhen is located, activated its top-level emergency response on this day, and all residents in Shenzhen were instructed not to leave their homes unless necessary."

3) Line 196: The authors mention that "observed at different heights". Does the data presented in Figure 1(a)-(c) represent the average value of vertical observations? Please clarify.

Yes, the pollutant concentration curves in Figure 2 are the average values for the whole vertical layer of observation. We have clarified it in the revised manuscript.

4) Line 205-209: I suggest removing these contents since the MDA8 ozone is a well-known indicator that is used to infer the magnitude of ozone pollution.

In order to maintain the integrity of the paper, we intend to retain the explanation of MDA8O3, while we have simplified it as follow.

"MDA8O3 is defined as the maximum 8-h moving average ozone concentration in a natural day, which is generally used to assess the severity of  $O_3$  pollution."

5) Line 210: Please clarify the potential reasons for this phenomenon. Possibly attributed to less titration effects of NO because primary NOx emission substantially decreased.

Yes, we agree with the comment. The higher average  $O_3$  during the lockdown period might be attributed to the low NO at night, therefore, although the daytime average  $O_3$  during the lockdown is much lower than that during the pre-lockdown, the daily average during lockdown is higher than that during the pre-lockdown. In the revised manuscript we emphasized the titration effects of NOx as follow.

"The ratios were generally greater than 1.0 in night, which clearly demonstrated less effective  $NO_x$  titration at night which leads to relatively higher ozone concentrations at night compared to pre-lockdown period."

6) Line 216-219: Please clarify the reason for the comparison of air pollutants and meteorological parameters between 2017 and 2020. Does the meteorological condition quite similar? Otherwise, it is not comparable.

The large scale circulations in the two periods were quite similar and the dominant wind direction were same, while the ground observed meteorological parameters were somehow different for stronger cold fronts in December 2017. The reason why the data in 2017 is included in the current study is that data in 2017 would provide more information on the long-term changes in pollutant concentration, emissions and climate.

7) Line 257-259: While the observations from the tower depict insignificant variations in meteorological parameters, the mesoscale process and large-scale synoptic pattern could still alter the air pollutants levels. I don't think the evidence is sufficient to make the conclusion.

Actually, in light of the classic theory on atmospheric diffusion, most of the influences of mesoscale or large-scale synoptic pattern on the pollutant concentrations are realized by the meteorological elements within boundary layer, such as wind speed, wind direction, relative humidity and temperature stratification, which determine the diffusion and transportation capability of the atmosphere (Pasquil, 1961; Pasquil, 1978; Gifford, 1961). Thus the comparison of the observed meteorological elements at the tower base during different period can used to determine whether the diffusion condition is similar or not. In response to your comments, we have added some information on the synoptic pattern analysis during the two periods, to further support our conclusion. We put the surface weather maps (1 map every 5 days) used to analyze the synoptic pattern below, from which it can be found that during the period of study, Shenzhen was primarily controlled by uniform pressure or weak high pressure ridge with sparse ground isobaric lines for most of the study period, which were quite typical circulation pattern in this area in winter. At the same time, We have weakened the tone on judging whether the weather is the major reason leading to the sudden change of the pollutant concentrations and have changed the word "impossible" into "unlikely".

2019-12-26

2019-12-31

---

## Author Comment (AC3)

**Reviewer #3**

This manuscript reports the vertical distributions of $PM_{2.5}$, $NO_x$ and $O_3$ at a tower in Pearl River Delta, China, before and during the COVID-19 lockdown, and analyzed the variations of these pollutants from different aspects and tried to give the responsible reasons. The authors finally concluded that the reductions of anthropogenic emissions by the lockdown were effective in mitigating both $PM_{2.5}$ and $O_3$. Although the datasets provided in this manuscript are unique and interesting, the organization and analysis is not sound and scientific. Many explanations for the observed data trends were too arbitrary, with little or even no evidence. Importantly, the role of meteorology on the observed trends was underestimated or even neglected. In addition, despite no $PM_{2.5}$ composition data in this manuscript, a large amount of papers on $PM_{2.5}$ in the PRD were not cited to support the data analysis. In my point of view, this is a very primary manuscript and no solid science has been drawn. It still has a long distance to publication at a high-level journal like ACP or similar journals, and thus I recommend rejecting it.

Thanks for the comments. This manuscript reports observed changes of vertical distribution of 3 kinds of pollutants ($PM_{2.5}$, $NO_x$ and $O_3$) by a 350m-tower in Pearl River Delta before and after the COVID19-induced lockdown. As far as we learn, this is possibly the first report based on tower observation in the massive studies on atmospheric environmental impact caused by COVID-19. We believe that these results will help to improve the understanding on the impact of human activities on the vertical distribution of pollutants, which is the most important scientific contribution of the current study. In the revised manuscript, we have added more discussion on the effect of meteorological elements.

Major comments:

1. There seem to be more pollutants which are routinely monitored in China's air quality network, such as SO2, PM10 and CO. Why these data are not included? They can provide more useful information on primary emissions. In addition, I don't understand why the authors included the 2017 data. As the authors said, both the emissions and weather in 2017 were different from those in 2020.

This study primarily focuses on the variation characteristics of pollutants in the vertical direction, and there is no observation of SO2 and CO on the SZMGT, which is not one of the state-controlled environmental monitoring stations, thus these two pollutants are not included in the current manuscript. For particulate matters, during the last decade, $PM_{2.5}$ has been the focus of a large number of studies on air quality in PRD area, while PM10 has not been the focus of attention for a long time(such as Yang et al., 2020; Huang et al., 2018; Zhou et al., 2016), so in the current study PM10 has not been analyzed either, since they cannot bring any new knowledge. The reason for adding the data in 2017 in the current manuscript is that we want to provide more information on the long-term changes in pollutant concentration.

**References:**

Yang, W., Chen, H., Wu, J., Wang, W., Zheng, J., Chen, D., Li, J., Tang, X., Wang, Z., Zhu, L., Wang, W..Characteristics of the source apportionment of primary and secondary inorganic $PM_{2.5}$ in the Pearl River Delta region during 2015 by numerical modeling. Environmental Pollution, 267: 115418. 2020.

Huang, Y., Deng, T., Li, Z., Wang, N., Yin, C., Wang, S., Fan, S..Numerical simulations for the sources apportionment and control strategies of $PM_{2.5}$ over Pearl River Delta, China, part I: Inventory and $PM_{2.5}$ sources apportionment. Science of The Total Environment, 634: 1631-1644. 2018.

Zhou, J., Xing, Z., Deng, J., Du, K..Characterizing and sourcing ambient $PM_{2.5}$ over key emission regions in China I: Water-soluble ions and carbonaceous fractions. Atmospheric Environment, 135: 20-30. 2016

2. Since meteorology largely influences or even dominates the variations of air pollutants, analysis of major meteorological parameters must always accompany the explorations of the variations of air pollutants. The comparison of averages of meteorological elements in Table 1 is far less than enough. Line 251 "it can be concluded that the meteorological conditions of the Shenzhen region were largely identical before and during the lockdown", this is too arbitrary. Since the vertical profiles/diurnal variations of the air pollutants in this study are not large (in Fig. 7), the simultaneous variations of temperature, RH and wind should be seriously analyzed. Unfortunately, meteorological analysis seemed to be totally forgotten starting from Section 3.2.

As explained in the response to the comments from reviewer 2, the influence of large-scale circulation conditions on near surface pollutant transport and diffusion will eventually be reflected in the meteorological elements in the boundary layer. Therefore, if the

statistical values of ground meteorological elements in the two stages were similar, it can be deduced that the impacts of climate on pollution in the two stages wee similar. Actually, beside the ground meteorological elements, the large scale circulation patterns were also similar in the two stages, which have been shown in our response to the 7[th] major comment from reviewer 2 and have been emphasized in the revised manuscript. In order to support the conclusion on "the meteorological conditions of the Shenzhen region were largely identical before and during the lockdown", two new figures are provided in the revised manuscript, as shown in Fig. R2. Fig. R2 was drawn by using TrajStat developed by Wang et al. (2009), which showed that the spatial distribution of the potential $PM_{2.5}$ source areas of Shiyan base were quite similar in the two different periods, which once again proved that the impacts of meteorology (especially the wind) on the air quality of Shiyan base were similar in pre-lockdown and during-lockdown periods. Furthermore, the comparison of the diurnal variation of wind speed, air temperature and wind speed was also made, which can be seen in Fig. R3, though they won't appear in the revised manuscript to avoid too many figures. Fig. R3 also shows that the the meteorological conditions of the Shenzhen region were generally identical before and during the lockdown on the diurnal variation scale.

[Figure]

Figure R2    Potential $PM_{2.5}$ source area for Shiyan base (a. pre-lockdown period; b. during lockdown period; Different colors indicate the probability that the airflow affecting the $PM_{2.5}$ concentration of Shiyan base passes through that area.)

[Figure]

Pre-CLD                        CLD

Figure R3    Comparison of the diurnal variation of wind speed, air temperature and relative humidity

While on the other hand, meteorological conditions did have influences on the variation of pollutants concentrations day by day. Thus, in the revised manuscript, we have added more discussion on the effect of meteorological conditions on pollutants concentrations in sections 3.1 and 3.2, though the variations day by day is not the concern of the current study. We also have added figures describing the diurnal variation of mixing layer height, which could help understand the diurnal curves of pollutant concentrations.

**References:**

Wang Y., Zhang X. Y., Draxler R. R..TrajStat: GIS-based software that uses various trajectory statistical analysis methods to identify potential sources from long-term air pollution measurement data. Environmental Modelling & Software, 24(8): 938-939. 2009.

3.  Section 3.3. Why the authors did not include the ground data in this section? The current vertical trends were represented by only three altitudes. If you add ground data in Fig. 7, will the vertical trends be the same?

As stated in the manuscript, the ground observation data are influenced by the surrounding trees and too local, which is why we did not include the ground level data in Fig. 7. Here we have added it in the revised manuscript (see Fig. R4), after adding the ground data in the figure, we found the trend changed. For the average case, the $PM_{2.5}$ and $NO_x$ are lower at the ground layer than that at 120m (influenced by the surrounding litchi trees), and the peak occur at 120 m in vertical scale, while an overall increasing trend with the increase in height for $O_3$ and $O_x$.

[Figure]

Figure R4 Comparison of the vertical distribution of pollutants

4.  There are too many subjective inferences without support or citation in the data analysis, such as:

Thanks for point out the inadequate and subjective discussions. For most of the contents you pointed out, we were trying to give possible reasons leading to the variation characteristics observed by the tower, and some of the explanations were based on conjectures and need further confirmation. We have strengthened the analysis based on further reference investigation and more information on meteorological conditions. At the same time, we will change the way of the statements to avoid arbitrary conclusions.

Line 267: "During night and early morning, the height of the mixing layer top is between 120m and 220m, so the curves of the upper and lower layers are quite different." Any evidence for the height of the mixing layer top?

Here we were trying to explain why the diurnal curve at 110-120m was different from that at 210-220m. We emphasized that this "probably reflects the uplift process of the mixing layer top in this area". Unfortunately, in the period of research in 2020, we do not

have related data to calculate the mixing layer height to support the conjecture (see Fig. R5). While, we have collected the LIDAR data during December, 2017, to calculate the mixing layer height (MLH) to support our conjecture based on method put forward by Morille et al. (2007).Though the LIDAR was out of service during the period of current study, the LIDAR data in the similar season of 2017 can still be used as a reference to learn the variation of MLH over Shiyan base. Figure R3 below shows the diurnal variation of MLH and the probability of MLH between 110 m and 220 m in December, 2017, which support our conjecture quite well.

[Figure]

(a)                                                                 (b)

Figure R5    Mixing layer height retrieved from LIDAR data in winter of Shiyan base (a. horizontal lines for average value and vertical lines for variation range; b. probability of MLH between 110 m and 220 m)

**References:**

Morille, Y., M. Haeffelin, P. Drobinski, et al.. STRAT: An automated algorithm to retrieve thevertical structure of the atmosphere from single-channel lidar data. Journal of Atmospheric and OceanicTechnology, 24, 761-775. 2007.

Line 270: "This may have been caused by the presence of dense forests near the ground observation point (Shiyan Base), which may have obstructed the dispersal of particulate matter and thus reduced the apparent $PM_{2.5}$ concentration." I think wind vertical profile may also influence it.

We are sorry for not being able to figure out how the vertical wind profile (see Fig. R6 below) could make the concentrations of pollutants at the altitude of 110-120m higher than those on the ground, since it is in a typical exponential wind profile form during the period of study.

[Figure]

Figure R6    Vertical profile of average wind speed during the period of study

Line 275: "High-height $PM_{2.5}$ is formed predominantly by chemical reactions, whereas low-height $PM_{2.5}$ may be derived from multiple sources (predominantly surface-level primary emissions)." This sentence is not sound. In the literature, secondary aerosols account for the major part of $PM_{2.5}$ in PRD even at the ground level. The influence of regional transport of secondary aerosols at

different altitudes was not discussed here.

Here, we do not deny the importance of secondary aerosol in ground level PM$_{2.5}$. We just want to emphasize that the impact of primary emission on the ground level PM$_{2.5}$ may be much higher than that in the air, which may explain the phenomenon observed by the tower. As for regional transportation, we analyzed the potential PM$_{2.5}$ source area for different height of SZMGT, and found that the potential source areas for all heights were generally similar to each other (see Fig. R7).

[Figure]

Figure R7 Potential PM$_{2.5}$ source area for different heights of Shiyan base (a. 110 m; b. 220m; c. 320 m.)

Line 288: "It is likely that the morning peak of the pre-lockdown curve was caused by direct emissions from nearby human activities. These emissions were therefore greatly reduced by the lockdown-mediated decrease in human activity and were more easily blocked by the dense forest around the ground observation point." Such analysis is too subjective.

We deleted it.

Line 312: "Since NO$_x$ is a primary pollutant, its significantly lower concentrations at the low altitudes implies that near-ground chemical reactions consume it more rapidly than the high-altitude chemical reactions do." What reactions? Why do they consume NO$_x$ more rapidly?

Here we double checked Figure 7, we found the absolute decrease of NO$_x$ at surface layer is much larger than those at the two layers above 200 m, which demonstrated that the surface layer is much more easily affected by emission change of NO$_x$, rather than reactions we proposed before.

Line 333: "A possible reason leading to this phenomenon is that in the area where the SZMGT is located, the key height of night chemical reactions may be around 110-120 m." Such analysis is irresponsible.

Deleted accordingly.

Lines 357-370. The discussion here is generally weak. How about PM$_{2.5}$ at the ground? What is the role of wind profile and/or regional transport? The authors may firstly check whether all vertical differences are statistically significant, especially when considering the measurement accuracy. "At the middle level and above, PM$_{2.5}$ is formed mainly by photochemical reactions (Li et al. 2020)", such citation is invalid. The vertical profile in Taiwan could be totally different and cannot support the discussion.

The reason why the ground level PM$_{2.5}$ concentration is not discussed here is that the ground level observation is influenced by the surrounding trees, as mentioned before. Comparing the ground level concentration with the concentrations of high levels above the trees may bring misleading information. For regional transportation, please refer to Fig. R7. In order to show possible influences of regional transportation on pollutant concentrations at different heights in the vertical direction, Fig. R7 provides the distribution of potential source areas for different heights obtained by a statistic analysis based on 72h backward trajectory data during the current study period. It can be seen from the figure that the distribution of potential source areas at different heights are generally similar, which are all located in the northeast of SZMGT. These areas cover the east part of the Pearl River Estuary, the urban belt along the southeast coast of China and the mountains of Nanling in northern Guangdong. These potential source areas may bring industrial emissions, traffic emissions from economically developed areas and biological VOCs from mountainous areas. However, because the distribution of potential source areas is generally consistent at all heights, it is difficult to judge the possible impact of regional transportation on the vertical distribution of pollutants.

We have removed the reference of Li et al (2020), but we still wish to cite the vertical observation results of VOCs observed in Kaohsiung, Taiwan because we believe it does have certain reference significance. Although there is a distance of about 660km

between Kaohsiung and Shenzhen, they are both located in the subtropical monsoon climate region and both belong to areas with developed industry and transportation. Furthermore, Kaohsiung is the most adjacent site to Shenzhen that has provided vertical VOCs distribution in surface layer. Since there is no other observation data available to support our discussion, the data in Kaohsiung cannot be neglected, for they raise at least a possibility that the photochemical reaction precursors at some particular height could be more abundant than at other heights, which is useful in supporting to explain what we observed on SZMGT.

Line 413: "It may be inferred that prior to the lockdown, $PM_{2.5}$ and $O_3$ did not have related sources. However, during the lockdown, both were likely to have a similar source." Such discussion is too casual.

Thanks for the suggestion, we deleted the casual statement.

Line 423: "At lower heights, a considerable part of $PM_{2.5}$ is primary source and had nothing to do with photochemical reactions", such analysis is irresponsible.

We changed as follow.

"At lower height, the primary source may have a higher contribution to the $PM_{2.5}$."

Line 431: "the primary aerosol like black carbon is not reduced," where are the black carbon data?

This sentence is deleted.

Minor comments:

Lines 272-273: what are the reasons for the peak occurring at 17:00–19:00?

It's an interesting question, actually we have no idea about it, we infer that the higher $PM_{2.5}$ at 220-325 m due to the primary emissions of $PM_{2.5}$ from the highway near the site, and it may not influence the PM level near the ground.

Line 298: a typo for mush.

Corrected accordingly.

Line 326: the high value can last to about 18:00, not only for 8:00 to 10:00.

Corrected accordingly.

Lines 347-349: how to calculate nitrate radical production rate?

We added the computational formula in the revised manuscript.

"We also checked the nitrate radical production rate (= $k_{NO2+O3}[NO_2][O_3]$) during the nighttime"

Lines 406-411: such basic description is not necessary.

We intend to keep this part to ensure the integrity of the paper.

Figures 9 and 10: the correlations were also affected by the data range and amount.

Yes, here we used all the data introduced in Section 2.

Lines 462-464: this sentence contradicts itself and should be rewritten.

Thanks, we rewrite it as follow.

"$PM_{2.5}$ became to be closely correlated to $O_3$, but not to $NO_x$, indicating that the formation of $PM_{2.5}$ during the lockdown might primarily be limited by atmospheric oxidants such as $O_3$."